# DIFFERENCE-AWARE VISIOLINGUISTIC REGULARIZATION FOR IMAGE CHANGE CAPTIONING

## ABSTRACT

Image Change Captioning (ICC) has emerged as an important task in multi-modal generative AI, aiming to generate natural language descriptions that reflect the differences between two similar images. Unlike traditional image captioning, ICC requires strong cross-image difference reasoning and language generation capabilities to handle diverse and complex scenarios. Recent advances have introduced MLLM-based methods for ICC, achieving impressive results. However, these approaches rely solely on caption-level supervision to implicitly infer and describe changes, which often results in the omission of fine-grained differences and suboptimal caption quality. To address this, we propose a Difference-Aware Visiolinguistic Regularization (DAVIR) paradigm that jointly regularizes the fine-tuning of MLLM from both visual and linguistic perspectives, enabling better adaptation to ICC. Specifically, we first introduce a fine-grained attention control module to regularize the final-layer self-attention maps of the MLLM's encoder, guiding it to focus on subtle changes during feature extraction. Second, we propose an entity prompt construction scheme to guide the MLLM's decoder and enhance caption generation quality. Extensive experiments on three benchmark datasets across different scenarios demonstrate that our method achieves state-of-the-art performance. The code will be released publicly.

## 1 INTRODUCTION

Image change captioning (ICC) (Hu et al., 2025; Li et al., 2025) aims to describe the semantic differences between two similar images in natural language. ICC plays an essential role in applications such as remote sensing monitoring (Liu et al., 2023a), surveillance analysis (Jhamtani & Berg-Kirkpatrick, 2018), and interactive image editing (Black et al., 2024). Compared with traditional image captioning (Kim et al., 2025; Zeng et al., 2025), ICC requires the model not only to understand each image individually but also to perform fine-grained cross-image reasoning.

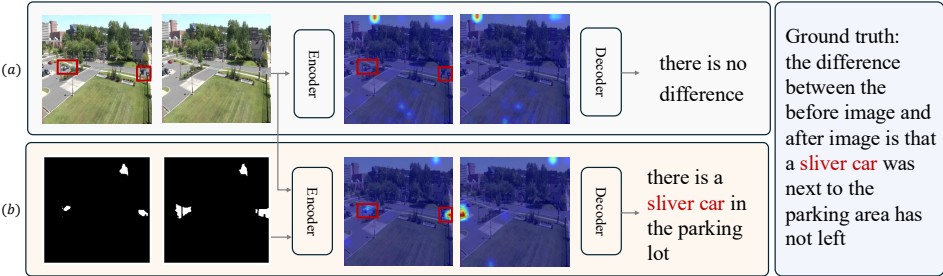

Figure 1: Examples of image change captioning in a video surveillance scenario, where the changes are usually subtle. (a) shows change localization results from the encoder of directly fine-tuned BLIP; (b) shows the results from our encoder fine-tuned with visual change masks generated by the customized clustering-based segmentation pipeline. We also visualize the ground-truth caption and generated captions by the directly fine-tuned BLIP and our method. The red words denote the names of the changed objects and the red boxes indicate changed objects.

Mainstream methods (Kim et al., 2021b; Yao et al., 2022; Tu et al., 2024a;c) typically begin by extracting visual features from each image using pre-trained backbones such as ResNet (He et al., 2016). They then model the difference features through either feature subtraction (Park et al., 2019) or matching paradigms (Shi et al., 2020), and subsequently feed the difference features into a trainable LSTM/Transformer-based decoder for caption generation. While these lightweight approaches gain encouraging performance, they often struggle with limited semantic reasoning and inadequate generalization in language generation, especially in complex or open-domain scenarios. With the emergence of multi-modal large language models (MLLMs), recent works (Zhang et al., 2024; Evennou et al., 2025) have begun adapting pre-trained MLLMs (e.g., BLIP-2 (Li et al., 2023) and InstructBLIP (Dai et al., 2023)) to ICC task via fine-tuning strategies. Benefiting from the powerful reasoning and generation capabilities of MLLMs, these approaches achieve impressive results on several well-known ICC benchmarks.

However, a major limitation exists in the aforementioned MLLM-based methods is: most of them rely solely on caption-level supervision to implicitly guide in learning visual difference representations. As illustrated in Fig.1(a), the visual encoders of these approaches often attend to irrelevant background regions rather than focusing on the objects, which may lead to the oversight of fine-grained changes and generate hallucinated descriptions that are semantically unrelated to the actual visual differences . An intuitive solution is to use pre-trained change detectors to generate masks of altered regions, which serve as pseudo-supervision for guiding the visual backbone in learning difference-related features. In fact, a few studies (Liu et al., 2024; Li et al., 2024) in remote sensing have adopted this strategy. However, unlike these works that focus on remote sensing, where numerous off-the-shelf detectors are readily available, we target ICC in more diverse scenarios (e.g., image editing, video surveillance, etc.), where such pre-trained detectors are not available. Besides, without explicit change-related text prompting, MLLMs are susceptible to hallucinations during inference, i.e., producing captions that are logically coherent yet semantically unrelated to the actual changes.

In this paper, we propose a novel **D**ifference-**A**ware **VI**siolinguistic **R**egularization (DAVIR) paradigm, which regularizes a pre-trained MLLM (i.e., InstructBLIP) using visual change masks and textual change entities during fine-tuning, thereby enabling better adaptation to the ICC task. Specifically, given a pair of images, we first compute a pixel-level difference map. Then, we design a fine-grained attention control (FAC) module, which applies a clustering-based segmentation pipeline to extract change masks for each image, highlighting regions likely to contain changes. FAC leverages these masks to regularize the final-layer self-attention maps of the InstructBLIP's encoder via a Kullback–Leibler divergence loss. By aligning attention maps with the change masks, FAC guides the encoder to focus more on change-relevant regions during feature extraction. Next, we incorporate a Q-Former to refine the features of the two images, ensuring that the most informative difference features are passed to the InstructBLIP's decoder.

To mitigate hallucination during caption generation, we devise an entity-prompt construction (EPC) scheme that uses textual entities representing potential changes to guide the decoding process. First, we extract candidate entities by identifying key noun phrases from ground-truth captions in the training set. These entities are then matched with potential changed regions derived from FAC, thus retrieving the most relevant entities for the image pair. Finally, we formulate an entity-centric instruction to prompt the decoder to focus on describing the actual changed objects.

Our main contributions are as follows:

- We propose a novel difference-aware visiolinguistic regularization (DAVIR) paradigm for ICC. Unlike prior works that rely solely on caption-level supervision, DAVIR jointly leverages visual change masks and textual change entities to regularize the fine-tuning of InstructBLIP, thereby enabling better adaptation to ICC.

- In DAVIR, FAC is designed to regularize the final-layer self-attention maps of the InstructBLIP's encoder, enabling it to focus more on subtle changes during feature extraction. Besides, EPC is devised to regularize the InstructBLIP's decoder to generate change captions that are both logically coherent and semantically accurate.

- DAVIR achieves state-of-the-art performance on three public change captioning benchmarks with different scenes, demonstrating the effectiveness and generic of our visiolinguistic regularization paradigm.

## 2 RELATED WORK

**Lightweight Methods.** Early works typically adopt a pre-trained ResNet (He et al., 2016) to encode each image in the image pair, then extract visual differences through using either feature subtraction (Park et al., 2019; Hosseinzadeh & Wang, 2021) or matching (Qiu et al., 2021; Huang et al., 2021). To enhance robustness under complex conditions such as illumination or viewpoint changes, some works (Kim et al., 2021a; Tu et al., 2021; Yue et al., 2023) design various attention-based interaction mechanism to predict common and difference features effectively . On the other hand, latest works (Tu et al., 2023c; 2024b; Li et al., 2025; Zhong et al., 2025) attempt to employ contrastive learning strategies to pull paired images closer in the feature space, aiming to reduce irrelevant variations. In addition, CLIP4IDC (Guo et al., 2022) attempts adapt a pre-trained CLIP model to ICC by introducing a cross-modal retrieval task, making the CLIP learn to extract difference-aware features. The aforementioned methods rely on trainable LSTM or Transformer-based decoders. However, due to the limited expressive capacity of these decoders, such lightweight approaches often struggle to generate high-quality change descriptions in real-world scenarios.

**MLLM-based Methods.** Recent works (Hu et al., 2024; Zhang et al., 2024; Evennou et al., 2025) introduce MLLMs, leveraging their powerful generation capabilities to produce more accurate change descriptions. Unlike previous methods that explicitly extract differences at the visual feature level, MLLM-based approaches identify changes implicitly via reasoning in the decoder. However, the encoder of MLLMs tends to capture global semantic information while overlooking fine-grained details. This bias toward global representations can result in insufficient attention to critical change regions. To address this issue, our method uses a Fine-Grained Attention Control (FAC) module, which guides the encoder to focus on the potential changed regions.

**Retrieval-Augmented Generation Methods.** We note that FINER-MLLM (Zhang et al., 2024) retrieves ground-truth captions from the training set to regularize the LLM's decoding. However, there are two differences between FINER-MLLM and our approach. First, FINER-MLLM performs cross-modal retrieval based on global features of image pairs and sentences. In contrast, our method adopts a more fine-grained strategy by using change regions identified by FAC and change entities extracted from ground-truth captions. Second, FINER-MLLM uses the retrieved tokens to regularize the predicted probability distribution of the decoder, whereas our approach explicitly uses the extracted change entities to guide the decoder during caption generation.

## 3 METHOD

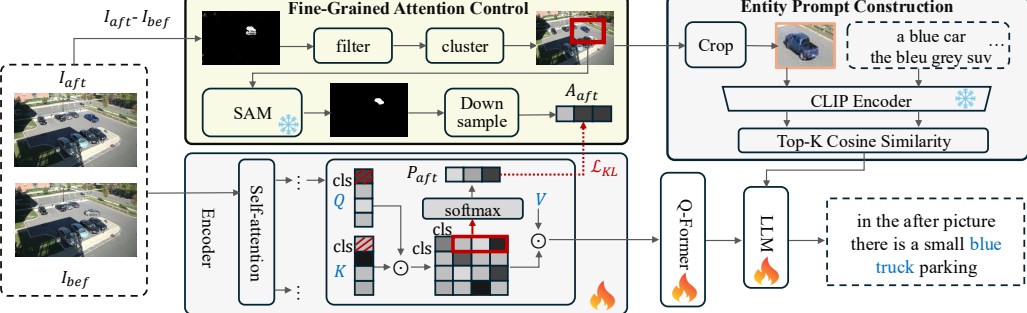

Figure 2: The overall architecture of the proposed method, which builds upon a pre-trained InstructBLIP model as the baseline MLLM. DAVIR comprises two core components: Fine-Grained Attention Control (FAC) and Entity Prompt Construction (EPC). Due to space limitations, we illustrate only the application of FAC in regularizing the final-layer self-attention map of InstructBLIP's encoder for the "after" image.

Given a pair of before and after images $(I_s, I_t)$, the goal of change captioning is to generate a natural language description $y$ that reflects the semantic changes between the two images. We formulate the task as learning a conditional generative model:

$$y = \mathcal{L}(I_s, I_t; \theta), \tag{1}$$

where $\mathcal{L}$ is a vision-language model parameterized by $\theta$. The overall architecture of our model is shown in Fig.2. In addition to the main components of the MLLM (ViT, Q Former, and LLM), we introduce two complementary components to enhance the model's ability to localize and describe changes: (1) Fine-Grained Attention Control, which regularizes the model's internal attention to align with the change region, and (2) Entity Prompt Construction, which supplements decoding with object-level text prompts extracted from visual differences.

## 3.1 CHANGE MASK EXTRACTION

To obtain precise localization of visual differences, we introduce an fine-grained mask extraction step based on image-level change detection. The goal is to provide the model with spatially grounded cues that highlight regions of semantic change while filtering out irrelevant variations. Given the before and after images $(I_s, I_t)$, we first compute a difference map to identify pixel-level discrepancies. To reduce noise and focus on salient changes, we apply thresholding and clustering:

$$\mathcal{C} = \text{DBSCAN}(\text{Diff}(I_s, I_t) > \delta), \tag{2}$$

where $\delta$ is a predefined threshold, and $\mathcal{C}$ denotes a set of spatially coherent clusters of changed pixels. Each cluster $c \in \mathcal{C}$ is then converted into a bounding box $\text{B}_c$ and center point $\text{C}_c$ to form a location prompt $P_c = (\text{B}_c, \text{C}_c)$. This prompt is fed into a segmentation model $\mathcal{S}$ (e.g., Segment Anything Model), which provides a high-quality mask prediction conditioned on the target image: $M_c = \mathcal{S}(I_t, P_c)$. To ensure that the predicted masks correspond to the originally detected changes, we compute the Intersection-over-Union (IoU) between each $M_c$ and the corresponding region $R_c$ from the difference map and only masks with sufficiently high IoU are retained and merged to form the final change mask:

$$M = \bigcup_{i:\text{IoU}_i \geq \tau} \mathcal{S}(I, p_i), \tag{3}$$

where $p_i$ is the prompt derived from the $i$-th cluster. This segmentation-driven mask $M$ not only suppresses irrelevant background noise but also provides accurate spatial guidance for downstream attention alignment and entity extraction, enhancing the model's ability to focus on true semantic changes.

## 3.2 FINE-GRAINED ATTENTION CONTROL

Given the extracted change mask $M \in \{0,1\}^{H \times W}$ from the previous step, which highlights regions of visual difference between $(I_s, I_t)$, we encourage the model to focus its attention on these regions during encoding. Both input images are first encoded using a shared vision backbone $\mathcal{E}$ (e.g., ViT):

$$F_{\text{bef}} = \mathcal{E}(I_{\text{bef}}), \quad F_{\text{aft}} = \mathcal{E}(I_{\text{aft}}), \tag{4}$$

where $F_{\text{bef}}, F_{\text{aft}} \in \mathbb{R}^{N \times D}$ represent the patch-wise visual tokens. We extract the self-attention maps from the last Transformer layer for each image:

$$A_{\text{bef}} = \text{Attn}(F_{\text{bef}}), \quad A_{\text{aft}} = \text{Attn}(F_{\text{aft}}), \tag{5}$$

where $A_{\text{bef}}, A_{\text{aft}} \in \mathbb{R}^{H_p \times W_p}$ correspond to spatial attention heatmaps reshaped from the [CLS]-to-patch attention. To align the model's attention with the actual change regions, we first downsample the binary mask $M$ to the patch resolution via average pooling, and then apply Gaussian blur to obtain a soft distribution. This is motivated by the observation that the original binary mask is often sparse and fragmented, which can make direct supervision unstable. The smoothing operation helps the model learn a more robust and continuous attention pattern:

$$\hat{M} = \text{GaussianBlur}(\text{Downsample}(M)), \tag{6}$$

where $\hat{M} \in [0,1]^{H_p \times W_p}$ is a soft mask. We supervise the attention maps via KL divergence:

$$\mathcal{L}_{\text{attn}} = \text{KL}(\sigma(A_{\text{bef}}) \parallel \sigma(\hat{M})) + \text{KL}(\sigma(A_{\text{aft}}) \parallel \sigma(\hat{M})), \tag{7}$$

where $\sigma(\cdot)$ denotes softmax normalization along spatial dimensions. This guides the model to focus on true change regions.

### 3.3 ENTITY PROMPT CONSTRUCTION

To further guide caption generation, we extract textual entities from the changed regions. We first apply the soft mask $\hat{M}$ to each image to obtain localized crops:

$$I_{\text{bef}}^m = I_{\text{bef}} \odot \hat{M}, \quad I_{\text{aft}}^m = I_{\text{aft}} \odot \hat{M}, \tag{8}$$

where $\odot$ denotes element-wise spatial masking. These cropped regions are passed to a pre-trained vision-language model $\mathcal{C}$ (e.g., CLIP) to retrieve top-$k$ matching text entities:

$$T = \mathcal{C}(I_{\text{bef}}^m) \cup \mathcal{C}(I_{\text{aft}}^m), \quad T = \{t_1, t_2, \ldots, t_k\}. \tag{9}$$

We convert these entity phrases into a textual prompt:

$$P = [\texttt{Prompt}] + [\texttt{Entity:}] + \text{Concat}(T), \tag{10}$$

and feed it along with visual features into a Q-former module $\mathcal{Q}$ to obtain aligned embeddings, which are then passed into the language model for caption generation:

$$y = \mathcal{L}(\mathcal{Q}(F_{\text{bef}}, F_{\text{aft}}), P). \tag{11}$$

### 3.4 JOINT TRAINING

Our model is jointly trained to generate change captions and align attention with the predicted change regions. Specifically, both before and after images are encoded using a vision encoder (e.g., ViT), and their patch features are passed through a shared Q-Former that performs cross-modal fusion with a pre-trained language model. During training, the model receives both visual inputs and optional entity-level textual prompts. The total training loss is defined as:

$$\mathcal{L}_{\text{total}} = \mathcal{L}_{\text{caption}} + \lambda \mathcal{L}_{\text{attn}}, \tag{12}$$

where $\mathcal{L}_{\text{caption}}$ is the standard cross-entropy loss over the generated caption tokens, and $\mathcal{L}_{\text{attn}}$ is the regularization loss that encourages the attention maps within the ViT to align with the extracted change mask. The hyperparameter $\lambda$ controls the strength of this attention supervision.

## 4 EXPERIMENTS

### 4.1 DATASETS

Our method is evaluated on three mainstream benchmarks: Spot-the-Diff (Jhamtani & Berg-Kirkpatrick, 2018), Image Editing Request (Tan et al., 2019), and LEVIR-CC (Liu et al., 2022). These datasets cover a range of domains, including video surveillance, remote sensing, and image editing, and encompass diverse types of changes such as object addition, removal, movement, and attribute modifications (e.g., color and texture).

**Spot-the-Diff** contains 13,192 real-world image pairs from video surveillance cameras, presenting challenges like subtle viewpoint and illumination variations.

**Image Editing Request** includes 4,900 image pairs accompanied by natural instructions for image edits, simulating user-guided change scenarios.

**LEVIR-CC** is a remote sensing dataset with 10,077 satellite image pairs describing large-scale land-use or construction changes, used to assess cross-domain robustness.

### 4.2 METRICS

We follow standard train/val/test splits for each dataset. All images are resized to $224 \times 224$ and tokenized using the InstructBLIP tokenizer. We report widely-used captioning metrics: BLEU-4 (Papineni et al., 2002), METEOR (Banerjee & Lavie, 2005), ROUGE-L (Lin, 2004), CIDEr (Vedantam et al., 2015), and SPICE (Anderson et al., 2016).

## 4.3 IMPLEMENTATION DETAILS

Our model builds on InstructBLIP, combining EVA-ViT-G, Q-Former, and Vicuna-7B with LoRA adapters applied to key modules and fine-tuned jointly, while other parameters remain frozen. We use SAM-ViT-H for pixel-level change masks and CLIP for region-text matching. Training details are provided in the appendix.

## 4.4 RESULTS AND COMPARISONS

We evaluate our method on three widely-used change captioning benchmarks: Spot-the-Diff, LEVIR-CC, Image-Editing-Request (IER). Our model is compared against both traditional vision-language methods and recent MLLM-based approaches. All models are evaluated using standard metrics, including BLEU, METEOR, ROUGE-L, CIDEr and SPICE.

Table 1: Performance comparison on the Spot-the-Diff dataset. $^\dagger$ denotes methods pretrained on a non-released large-scale dataset before fine-tuning on Spot-the-Diff. $^*$ denotes methods fine-tuned on Spot-the-Diff only.

| Method | B | M | R | C | S |
|---|---|---|---|---|---|
| DUDA+Aux (Hosseinzadeh & Wang, 2021) | 8.1 | 12.5 | 29.9 | 34.5 | – |
| VACC (Kim et al., 2021a) | 9.7 | 12.6 | 32.1 | 41.5 | – |
| BiDiff (Sun et al., 2022) | 6.6 | 10.6 | 29.5 | 42.2 | – |
| CLIP4IDC (Guo et al., 2022) | 11.6 | 14.2 | 35.0 | 47.4 | – |
| VARD (Tu et al., 2023a) | – | 12.5 | 29.3 | 30.3 | 17.3 |
| SCORER (Tu et al., 2023c) | 10.2 | 12.2 | – | 38.9 | 18.4 |
| DIRL+CCR (Tu et al., 2024a) | 10.3 | 13.8 | 32.8 | 40.9 | 19.9 |
| SMART (Tu et al., 2024b) | - | 13.5 | 31.6 | 39.4 | 19.0 |
| OneDiff$^\dagger$ (Hu et al., 2024) | 12.8 | 14.6 | 35.8 | 56.6 | – |
| FINER-MLLM$^*$ (Zhang et al., 2024) | **12.6** | 13.5 | 34.7 | 59.9 | 21.1 |
| BLIP2IDC$^*$ (Evennou et al., 2025) | 11.6 | 14.0 | 35.3 | 52.2 | – |
| **DAVIR** $^*$**(Ours)** | 12.1 | **15.8** | **35.9** | **61.9** | **23.0** |

**Results on Spot-the-Diff Dataset.** On the Spot-the-Diff dataset, our method achieves state-of-the-art performance across most of evaluation metrics, as shown in Table 1. This dataset features real-world image pairs with complex, human-written captions, making it particularly challenging. Traditional change captioning models, which rely on lightweight architectures and purely visual reasoning, struggle to handle such rich linguistic and semantic variance. As a result, recent works increasingly adopt MLLMs to improve semantic understanding and generation quality. Compared with these MLLM-based baselines, our method consistently outperforms them in METEOR, ROUGE-L, CIDEr and SPICE. This performance gain highlights the effectiveness of fine-grained attention regularization and the entity-guided textual prompting. By enhancing the alignment between visual attention and change regions, and by explicitly guiding the LLM with semantically relevant entity prompts, our model generates more accurate and descriptive captions. The significant improvements across all metrics demonstrate that our visiolinguistic strategy effectively leverages both visual and textual cues for fine-grained change understanding in real-world scenarios.

**Results on Image Editing Request Dataset.** Our method achieves state-of-the-art performance on the Image Editing Request (IER) dataset, particularly excelling in CIDEr and BLEU scores, as shown in Table 2. The IER dataset contains synthetic image pairs with subtle and localized edits, as well as background variations that increase the difficulty of precise change localization. Our approach effectively guides the model to focus on true regions of change through SAM-based difference masks and fine-grained attention alignment. Meanwhile, the entity-guided textual prompting enhances the model's semantic precision, enabling it to generate accurate and detailed captions even in the presence of distracting background differences. The combination of visual supervision and targeted language guidance allows our model to robustly distinguish between meaningful edits and irrelevant variations.

Table 2: Performance comparison on the IER dataset. * denotes methods fine-tuned on IER only.

| Method | B | M | R | C | S |
|---|---|---|---|---|---|
| DUDA (Park et al., 2019) | 6.5 | 12.4 | 37.3 | 22.8 | – |
| BiDiff (Sun et al., 2022) | 6.9 | 14.6 | 38.5 | 27.7 | – |
| CLIP4IDC (Guo et al., 2022) | 8.2 | 14.6 | 40.4 | 32.2 | – |
| NCT (Tu et al., 2023b) | 8.1 | 15.0 | 38.8 | 34.2 | 12.7 |
| VARD (Tu et al., 2023a) | 10.0 | 14.8 | 39.0 | 35.7 | – |
| SCORER (Tu et al., 2023c) | 10.0 | 15.0 | 39.6 | 33.4 | 12.6 |
| SMART (Tu et al., 2024b) | 10.5 | 15.2 | 39.1 | 37.8 | 13.0 |
| DIRL+CCR (Tu et al., 2024a) | 10.9 | 15.0 | 41.0 | 34.1 | 11.0 |
| BLIP2IDC* (Evennou et al., 2025) | 13.0 | 15.8 | 40.7 | 55.3 | 13.6 |
| FINER-MLLM* (Zhang et al., 2024) | 12.9 | 15.1 | 40.4 | 52.1 | 14.4 |
| **DAVIR*(Ours)** | **16.7** | **16.2** | **41.2** | **65.2** | **16.9** |

**Results on LEVIR-CC Dataset.** As shown in Table 3, our method achieves state-of-the-art performance on the LEVIR-CC dataset, outperforming existing approaches across all metrics. This result highlights the effectiveness of our fine-grained attention alignment strategy in handling complex change scenarios. LEVIR-CC contains numerous subtle and localized changes, making it a challenging benchmark for change captioning. By introducing SAM-based difference masks to guide the ViT's attention distribution and explicitly prompting the language model with matched entity descriptions, our method is better able to identify and describe the true semantic changes while filtering out background noise and viewpoint differences. These results suggest that fine-grained visual supervision and entity-aware prompting are crucial for accurate and grounded change description in fine-resolution remote sensing imagery.

Table 3: Performance comparison on the LEVIR-CC dataset. [†] denotes methods pretrained on a non-released large-scale dataset before fine-tuning on LEVIR-CC. * denotes methods fine-tuned on LEVIR-CC only.

| Method | B | M | R | C |
|---|---|---|---|---|
| DUDA (Park et al., 2019) | 57.8 | 37.2 | 71.0 | 124.3 |
| MCCFormers-D (Qiu et al., 2021) | 56.4 | 37.3 | 70.3 | 124.4 |
| MCCFormers-S (Qiu et al., 2021) | 56.7 | 36.2 | 69.5 | 120.4 |
| RSICCformer (Liu et al., 2022) | 62.8 | 39.6 | 74.1 | 134.1 |
| Chg2cap (Chang & Ghamisi, 2023) | 64.5 | 40.0 | 75.1 | 136.6 |
| PSNet (Liu et al., 2023a) | 62.1 | 38.8 | 73.6 | 132.6 |
| CARD (Tu et al., 2024c) | 65.4 | 40.0 | 74.6 | 137.9 |
| RDD+ACR (Li et al., 2025) | **65.6** | 40.3 | 75.5 | 138.3 |
| Change3D (Zhu et al., 2025) | 64.4 | 40.0 | 75.1 | 138.3 |
| SEN[†] (Zhou et al., 2024) | 64.1 | 39.6 | 74.6 | 136.0 |
| Prompt-CC* (Liu et al., 2023b) | 63.5 | 38.8 | 73.7 | 136.4 |
| **DAVIR*(Ours)** | 65.1 | **40.9** | **75.9** | **140.7** |

## 4.5 ABLATION STUDY

**Ablation Study on each module.** To assess the contribution of each proposed component, we conduct a comprehensive ablation study by incrementally removing or disabling individual modules: (1) the SAM-guided attention regularization loss, (2) the entity prompt construction mechanism. As shown in Table 4, removing any of these modules leads to a clear performance drop across all evaluation metrics, confirming their complementary benefits.

Table 4: Ablation study on each modules on Spot-the-Diff dataset.

| FAC | EPC | B | M | R | C | S |
|---|---|---|---|---|---|---|
| ✗ | ✗ | 11.5 | 13.0 | 33.7 | 60.4 | 21.6 |
| ✓ | ✗ | 12.0 | 13.2 | 34.1 | **62.4** | 22.2 |
| ✗ | ✓ | 11.9 | **16.4** | 35.2 | 58.4 | 22.9 |
| ✓ | ✓ | **12.1** | 15.8 | **35.9** | 61.9 | **23.0** |

Specifically, disabling the regularization loss causes the largest degradation in CIDEr and SPICE, highlighting the importance of accurate visual grounding. Removing entity prompt construction reduces the model's ability to produce semantically rich and precise descriptions, especially in complex scenarios. Lastly, bypassing CLIP-based filtering and using randomly selected entities introduces irrelevant information, resulting in reduced caption quality. These results collectively validate the effectiveness of our full pipeline design, where both visual alignment and semantic guidance are essential for fine-grained and grounded change captioning.

**Ablation study on entity num.** We conduct ablation studies on the number of entities used in our entity-guided prompting strategy to investigate the optimal level of guidance information. Table 5 shows the performance with different numbers of entities extracted from the retrieval corpus. Our results demonstrate that more entities do not necessarily lead to better performance. The optimal performance is achieved with a single entity, while using four entities results in the lowest scores. This

Table 5: Effect of different entity number on Spot-the-Diff dataset.

| Entities | B | M | R | C | S |
|---|---|---|---|---|---|
| 1 | **12.1** | 15.8 | **35.9** | 61.9 | **23.0** |
| 2 | 10.5 | 14.9 | 34.5 | **62.3** | 22.1 |
| 3 | 11.8 | **16.2** | 33.8 | 60.2 | 22.5 |
| 4 | 11.2 | 15.3 | 35.2 | 61.0 | 21.4 |

phenomenon occurs because additional entities introduce noise and irrelevant information that can mislead the model's attention and generation process. While appropriate prompting with focused entity guidance effectively directs the model toward relevant change subjects and improves description accuracy, excessive prompting creates conflicting signals that negatively impact performance.

**Ablation study on regularization loss.** We investigate different loss functions for supervising the attention alignment between vision transformer attention weights and masks. Table 6 compares the performance of various regularization loss formulations. The results demonstrate that KL divergence achieves the best performance across all metrics, significantly outperforming the baseline without alignment supervision and other loss formulations. This superiority stems from the fact that both attention weights and

Table 6: Effect of different regularization loss on Spot-the-Diff dataset.

| Loss | B | M | R | C | S |
|---|---|---|---|---|---|
| None | 11.9 | **16.4** | 35.2 | 58.4 | 22.9 |
| MSE | 11.1 | 15.8 | 34.9 | 58.2 | 22.7 |
| CE | 11.6 | 16.1 | 35.5 | 58.5 | **23.1** |
| KL | **12.1** | 15.8 | **35.9** | 61.9 | 23.0 |

ground-truth masks can be naturally interpreted as probability distributions over spatial locations, making KL divergence the most appropriate choice for measuring distributional similarity. KL divergence effectively encourages the model to learn attention distributions that closely match the spatial distribution of actual changes.

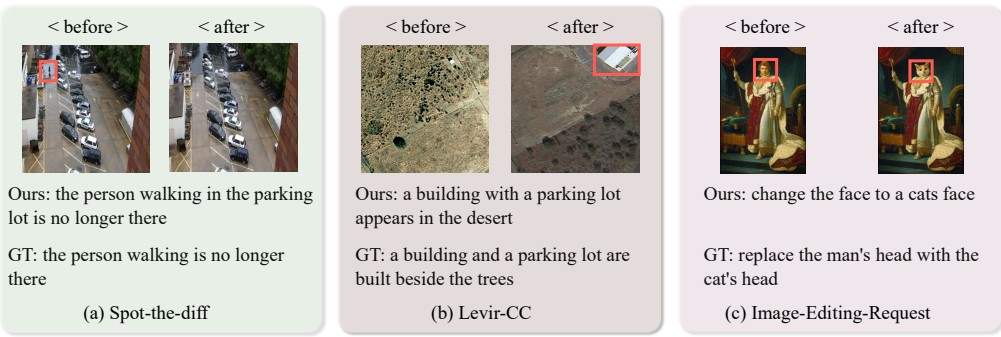

Figure 3: Examples of captions generated by our model on three datasets: (a) Spot-the-Diff, (b) LEVIR-CC, and (c) Image Editing Request. Each example shows before and after images, predicted captions, and the corresponding ground truth.

## 4.6 QUALITATIVE ANALYSIS

To further validate the effectiveness of our approach, we present qualitative comparisons of generated captions and visualization of attention mechanisms in Fig.3 and Fig.4.

**Caption Generation Comparison.** Fig.3 demonstrates representative examples comparing our method with baseline methods across diverse image editing scenarios. Our method consistently generates more accurate and detailed descriptions that precisely capture the nature of changes. This improvement stems from our entity-guided prompting mechanism that provides structured context about potential change subjects, enabling the model to focus on relevant objects and generate more semantically meaningful descriptions. The enhanced spatial understanding from mask regularization loss also contributes to more accurate localization descriptions, avoiding common errors like misidentifying change locations or confusing foreground and background modifications.

**Attention Mechanism Visualization** Fig.4 illustrates the attention maps from the ViT's final layer, comparing our method with and without mask regularization loss supervision. The visualization clearly shows that our regularization loss constraint significantly improves attention focus on actual change regions. Without alignment supervision, attention weights are scattered across the entire image, often focusing on salient but irrelevant objects. In contrast, with our KL divergence-based regularization loss, attention gradually becomes concentrated on the specific regions where changes occur, such as added/removed objects or modified areas. This focused attention directly translates to better change understanding and more accurate caption generation, as the model's visual processing is explicitly guided toward the most relevant image regions. The improved attention localization validates our hypothesis that explicit spatial supervision enhances the model's ability to ground language generation in visually relevant content, leading to the substantial performance improvements observed in quantitative metrics.

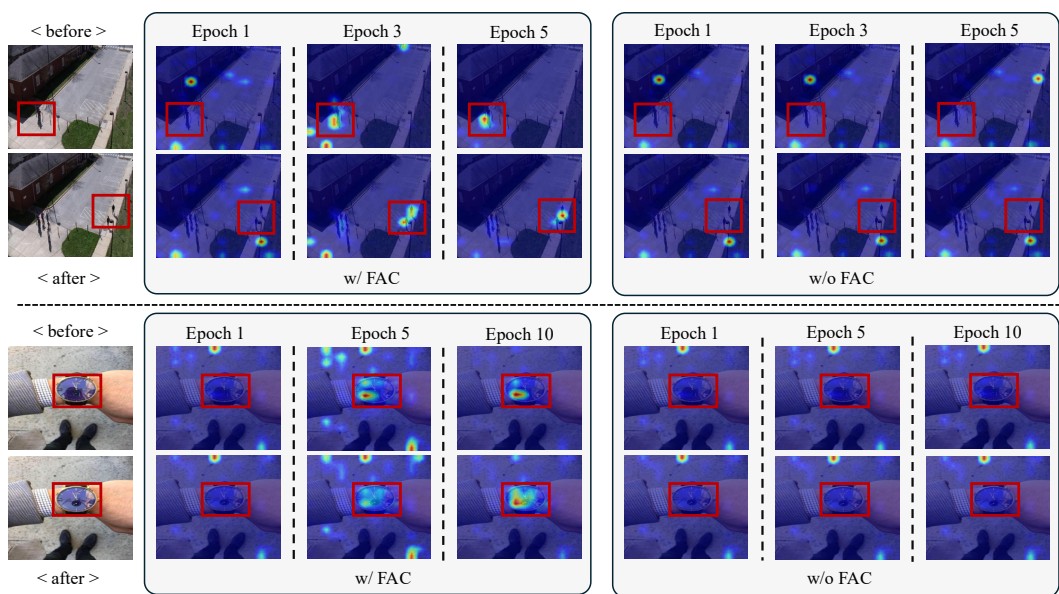

Figure 4: Visualization of attention map evolution across training epochs. The top row shows results on the Spot-the-Diff dataset, and the bottom row shows results on the IER dataset. The left part shows results from our model with the proposed FAC module, while the right column shows results from a baseline model without FAC (i.e., vanilla InstructBLIP).

# 5 CONCLUSION

In this paper, we propose a novel difference-aware visiolinguistic regularization (DAVIR) paradigm, which incorporates fine-grained attention control (FAC) module and entity prompt construction (EPC) scheme to facilitate the adaptation of InstructBLIP to the ICC task during fine-tuning. FAC first leverages a clustering-based segmentation pipeline to obtain masks of potential changes, and then uses these masks to guide the encoder of InstructBLIP to focus on actual change regions. During caption generation, EPC introduces change-related contextual cues to mitigate hallucination and improve the accuracy of generated descriptions. Extensive experiments demonstrate that our method achieves state-of-the-art performance on three challenging datasets.

ETHICS STATEMENT

This work does not raise any ethical concerns. It does not involve human subjects, sensitive data, or potentially harmful applications, and therefore no specific ethical considerations are applicable.

REPRODUCIBILITY STATEMENT

The code for running the experiments and for generating all visualizations can be accessed at the following anonymous link: https://github.com/x7p9QzLm2R/DAVIR.git. This includes scripts for training, evaluation, and visualization, ensuring that all results reported in the paper can be reproduced. Additional details for dataset preparation, model configurations, and experimental settings are provided in the code repository.

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

# APPENDIX

## A    LLM USAGE

We use LLM only as an auxiliary tool for language polishing and formatting adjustments. The LLM does not contribute to the research ideation, methodology design, experiments, analysis, or substantive writing of the paper.

## B  IMPLEMENTATION DETAILS

Our model is based on the InstructBLIP architecture, using EVA-ViT-G, Q-Former, and Vicuna-7B. We apply LoRA adapters with rank $r = 4$ to the QKV projections of EVA-ViT-G and Q-Former, and rank $r = 8$ for Vicuna-7B, and fine-tune them jointly. All other parameters remain frozen. We adopt SAM-ViT-H to obtain pixel-level change masks and use CLIP for region-text matching. We train for 20 epochs with a batch size of 16 and learning rate $1 \times 10^{-5}$ on a single NVIDIA A800 GPU. The KL loss weight is set to $\lambda = 0.1$.

## C  ADDITIONAL VISUALIZATION RESULTS

Fig.6 shows several attention map visualizations on the Spot-the-Diff dataset. Fig.5 shows several attention map visualizations on the IER dataset.

## D  BLIP-BASED VISION-LANGUAGE MODELS

BLIPLi et al. (2022) and its successor BLIP-2Li et al. (2023) propose a unified framework for connecting vision encoders with LLMs via a lightweight query transformer (Q-Former). These models support multi-modal instruction tuning and achieve strong performance on various vision-language tasks. InstructBLIPDai et al. (2023) further adapts this framework to instruction-following settings, allowing for flexible language-guided image understanding.

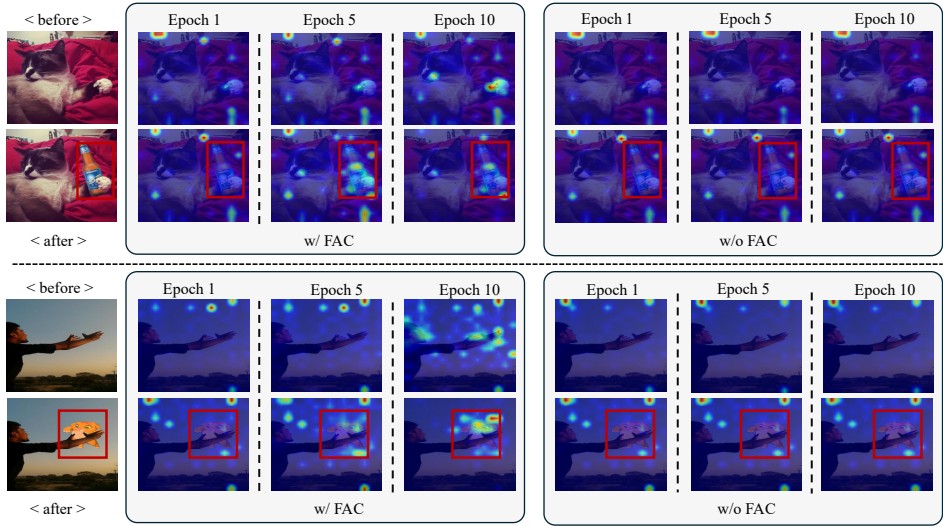

Figure 5: Additional visualization results on IER dataset.

## E  SEGMENT ANYTHING MODEL

Segment Anything Model (SAM)Kirillov et al. (2023) is a foundation model for segmentation that generalizes across domains with strong zero-shot performance. Given a prompt (point, box, or mask), SAM outputs high-quality instance-level segmentations. Due to its scalability and flexibility, SAM has been widely adopted for downstream tasks such as object discovery, visual grounding, and image editing. In our work, we use SAM to generate dense change masks, which provide spatial priors for both visual attention regularization and entity selection.

## F  ADDITIONAL ABLATION STUDIES

Table 7 shows the performance of our model under different values of the hyperparameter $\lambda$. We observe that setting $\lambda = 0.1$ achieves the best overall results across most metrics, including BLEU

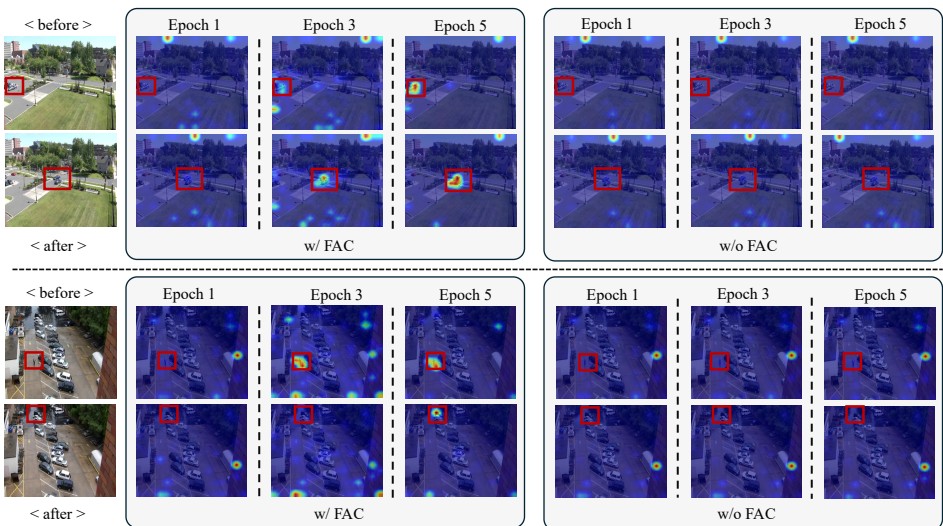

Figure 6: Additional visualization results on Spot-the-Diff dataset.

Table 7: Effect of different $\lambda$ values on Spot-the-Diff dataset.

| $\lambda$ | B | M | R | C | S |
|---|---|---|---|---|---|
| 0.1 | **12.1** | **15.8** | **35.8** | 61.9 | **23.0** |
| 0.2 | 11.7 | 15.6 | 35.2 | **62.4** | 22.6 |
| 0.3 | 10.9 | 15.1 | 34.0 | 60.1 | 21.8 |
| 0.4 | 11.1 | 15.3 | 34.4 | 59.7 | 21.5 |

(12.3), METEOR (16.5), ROUGE (36.1), and SPICE (23.3). Interestingly, while $\lambda = 0.2$ slightly underperforms in most metrics, it achieves the highest CIDEr score (62.4), suggesting that a moderate level of regularization can enhance consensus with reference captions. However, further increasing $\lambda$ to 0.3 and 0.4 leads to consistent performance drops, indicating that excessive regularization may hinder the model's ability to capture detailed semantic differences.

