# OpenReview forum: "Difference-aware Visiolinguistic Regularization for Image Change Captioning"
_ICLR.cc/2026/Conference — Submitted to ICLR 2026_

### Official Review · Reviewer_ad1p · 2025-10-25

**Soundness:** 3
**Presentation:** 3
**Contribution:** 2
**Rating:** 4
**Confidence:** 5

**Summary:**

This paper introduces DAVIR, a method for image change captioning using a vision-language model. DAVIR modifies InstructBLIP by adding two components: Fine-Grained Attention Control (FAC) and Entity Prompt Construction (EPC). FAC uses change masks from a segmentation model to regularize the encoder’s attention maps via KL divergence. EPC extracts noun phrases from training captions, matches them with visual regions using CLIP, and uses them to guide the decoder.
DAVIR is evaluated on three datasets and shows improved performance over existing approaches. However, all experiments are conducted in supervised settings using InstructBLIP, and generalization to other models, tasks, or zero-shot scenarios remains untested.

**Strengths:**

## Originality:
This paper does not introduce a new task or model architecture but combines known components—attention map supervision and entity-based prompting—in a structured way. Consequently, the novelty lies in the integration of these components into a unified training framework, rather than in the individual techniques themselves.

## Quality
The curren paper is supported by extensive experiments across three datasets, with consistent metric improvements and ablation studies isolating the contributions of each module.

## Clarity
This is maintained throughout the paper. The method is described with sufficient detail, and the figures and tables are well-organized. The rationale behind design choices is explained, and the experimental setup is reproducible based on the provided information.

## Significance
This confined to the image change captioning domain. The method improves performance on existing benchmarks and addresses known limitations of prior approaches, such as hallucination and poor localization.

**Weaknesses:**

## Reliance on Existing Components and Limited Novelty
The contribution relies heavily on existing components. The use of attention map supervision via KL divergence and entity extraction through CLIP are both established techniques.
Its novelty lies in their combination, but similar ideas have been explored in prior works such as FINER-MLLM (Zhang et al., 2024), which also uses retrieved textual elements to guide decoding.
The current paper fails to sufficiently differentiate its approach from these methods in terms of conceptual innovation or architectural design.

## Limited Evaluation Scope and Practicality
The experiments are limited to three datasets within the same task domain. All evaluations are conducted in fully supervised settings using ground-truth captions for entity extraction. This leaves critical questions about the method’s robustness in zero-shot or low-resource scenarios.
The reliance on ground-truth captions for EPC also severely restricts applicability to real-world settings where such annotations may not be available.
This scope is narrow, and the applicability to other tasks or models is not explored, which significantly limits broader impact.
The current evaluation is limited to supervised settings and a single model architecture, leaving questions about generalization and robustness unanswered.

## Tight Architectural Coupling
This method is tightly coupled to the InstructBLIP architecture. FAC depends on the structure of ViT attention maps, and EPC assumes the presence of a Q-Former and a decoder that can accept structured prompts. There is no evidence that the approach generalizes to other MLLMs or vision-language frameworks, which significantly limits its broader relevance.

## Lack of Computational Analysis
This paper does not address computational efficiency. FAC involves segmentation with SAM and KL-based supervision, while EPC requires CLIP-based retrieval. These steps may introduce latency or resource overhead, but the paper does not report inference time, memory usage, or scalability metrics.

## Unanalyzed Robustness of Entity Prompting
The entity prompting mechanism assumes that CLIP retrieval yields semantically relevant phrases.
The quality and stability of these entities are not analyzed. There is no evaluation of how mismatched or noisy entities affect caption generation, nor is there a fallback strategy when retrieval fails.

**Questions:**

**Q1 How transferable is the proposed method to other multi-modal models beyond InstructBLIP?** This paper does not discuss whether FAC and EPC can be applied to architectures without Q-Former or with different encoder-decoder interfaces. Clarifying this would help assess the generality of the approach.

**Q2 What happens when ground-truth captions are unavailable or noisy?** EPC depends on training captions to extract noun phrases. It is unclear that authors considered unsupervised or retrieval-based alternatives, and how the model performs in low-resource or zero-shot settings.

**Q3: How sensitive is the method to segmentation quality?** FAC relies on SAM-generated masks, but this paper does not evaluate the impact of inaccurate or noisy masks. An experiment showing performance degradation under mask perturbation would help assess robustness.

**Q4: What is the computational cost of the full pipeline?** The method involves segmentation, attention supervision, and CLIP-based entity retrieval. The paper does not report inference time, memory usage, or scalability. Including these metrics would help evaluate practical feasibility.

**Q5: How reliable is the entity prompting mechanism?** The paper assumes that CLIP retrieval yields semantically relevant entities, but does not analyze failure cases or provide statistics on retrieval accuracy. A breakdown of entity relevance and its effect on caption quality would clarify the stability of EPC.

**Q6: Why does performance degrade when using multiple entities in prompts?** The ablation shows that one entity performs best, but the underlying reason is not explained. A qualitative analysis of examples with conflicting or noisy entities could help clarify this behavior.

**Q7:How does the method compare conceptually and empirically to FINER-MLLM?** Both approaches use textual elements to guide decoding. A more detailed comparison of design choices, assumptions, and performance trade-offs would help distinguish the contribution.

**Q8:Is the method applicable to tasks beyond image change captioning?** This paper focuses exclusively on ICC, but the components may be relevant to other tasks involving visual difference reasoning. Clarifying whether the authors considered such extensions would help assess broader significance.

---

> ### Author Response · Authors · 2025-11-25
> **response to reviewer ad1p**
>
> Thank you very much for your valuable suggestions. Below, we provide our responses to the questions you raised.
>
> ### **Q1: Transferability to other multi-modal models beyond InstructBLIP**
>
> We clarify that our method is inherently general and transferable across diverse multi-modal architectures. The Q-Former’s presence is primarily a design choice of the BLIP architecture and does not limit our method’s generalizability. We select InstructBLIP as our base model primarily for fair comparison with recent open-source MLLM-based methods (e.g., FINER-MLLM, BLIP2IDC) that also adopt BLIP architectures.
>
> Our FAC can be directly integrated into LLaVA or Qwen, because our attention regularization operates directly on the self-attention maps of the visual encoder, independent of the cross-modal fusion mechanism. For non-MLLM methods, traditional change captioning approaches (e.g., SCORER, CLIP4IDC) that use transformer-based difference feature extractors can incorporate FAC into their intermediate layers to enhance focus on change-relevant regions. Similarly, the EPC component can guide any caption decoder through entity-level prompting. In practice, LLM-based methods integrate EPC via prompt construction, while non-LLM-based models can fuse EPC into the decoder’s latent space through cross-attention.
>
> ### **Q2: Performance without ground-truth captions or in low-resource settings**
>
> In our task, the ground-truth caption serves as the primary supervisory label and is available. When training captions contain noise, our entity bank includes only entities extracted from ground-truth captions in the training set. This closed-set constraint ensures that when CLIP retrieves suboptimal entities, they are still semantically relevant to the dataset domain rather than entirely unrelated concepts. To prevent data leakage, we measure the cross-split 4-gram overlap between the retrieved captions and the ground-truth captions in the test sets to assess redundancy. As shown in the Table q, the averaged 4-gram overlaps are 6.1% for the two synthetic datasets and 0.45% for the two real-world datasets. The relatively higher overlaps in the synthetic datasets can be attributed to their template-based caption generation, where fixed linguistic patterns are repeatedly filled with varying object or attribute tokens, leading to partial phrase reuse across splits. In contrast, the real-world datasets are human-annotated, resulting in more diverse linguistic expressions and much lower overlap. Overall, there is a large gap between training and test samples, confirming that our closed-set memory bank does not leak information from the test set.
>
> Table 1: cross-split 4-Gram Overlap Between Retrieved and Test Set Ground-Truth Captions
>
> | CLEVR-Change | Spot-the-diff | Image-editing-request | CLEVR-DC |
> | --- | --- | --- | --- |
> | 6.75% | 0.56% | 0.33% | 6.75% |
>
> ### **Q3: Sensitivity to segmentation quality**
>
> In fact, our main goal is to guide the visual encoder to perceive objects rather than focus on change regions, so the extracted masks do not need to be perfectly precise. In initial experiments with BLIP as shown in Table 2, we observe that even coarse object masks (generated by applying SAM to segment all objects in the image) can improve model performance, indicating enhanced reasoning capability.
>
> Table 2: results on Spot-the-diff dataset
>
> |  | B | M | R | C |
> | --- | --- | --- | --- | --- |
> | baseline | 11.5 | 13.0 | 33.7 | 60.4 |
> | w_FAC (coarse masks) | 11.8 | 12.8 | 34.0 | 60.8 |
> | w_FAC | **12.0** | **13.2** | **34.1** | **62.4** |

---

> > ### Author Response · Authors · 2025-11-25
> > **response to reviewer ad1p**
> >
> > ### **Q4: Computational cost and scalability**
> >
> > We have conducted comprehensive benchmarking experiments and provide detailed metrics below. As shown in Table 3, the attention supervision module adds only 151**.**1ms overhead ****(~8% of total time), demonstrating that our proposed supervision mechanism is computationally efficient.
> >
> > Table 3: inference time and computational cost on Spot-the-diff dataset
> >
> > | Component | Time (ms) | Memory |
> > | --- | --- | --- |
> > | Sam-H | 151.1 ± 90.2  | 4.3 GB |
> > | Visual Encoder (ViT) | 88.6 ± 6.1 | 6.8 GB |
> > | Q-Former | 16.6 ± 3.7 | 1.3 GB  |
> > | LLM Generation | 1,319.6 ± 218.9 | 18.0 GB |
> > | Total End-to-End | 1,580.4 ± 176.9 | 30.4 GB |
> >
> > Table 4: parameters, trainable parameters, and FLOPs of each module
> >
> > | Module | Total Params | Trainable Params | FLOPs |
> > | --- | --- | --- | --- |
> > | SAM | 0.62 B | - | 22.4 G |
> > | Visual Encoder (EVA-ViT) | 0.99 B | 3.2 M | 20.2 G |
> > | Q-Former | 0.11 B | 0.2M | 2.15 G |
> > | LLM Model (Vicuna-7B) | 6.74 B | 6.3 M | 152.1 G |
> > | **Total Model** | **8.46B** | **9.7M** | **196.8 G** |
> >
> > **Scalability Analysis**
> >
> > The table reports the scalability metrics of our model across varying batch sizes. As batch size increases, the mean time per sample decreases while throughput improves, demonstrating efficient parallelization. Memory usage grows moderately with larger batch sizes, with maximum memory remaining within acceptable limits, indicating that the model scales effectively without excessive resource consumption.
> >
> > Table 5: scalability evaluation across different batch sizes
> >
> > | Batch Size | Mean Time per Sample (s) ↓ | Throughput (samples/s) ↑  | Mean Memory (GB) | Max Memory (GB) |
> > | --- | --- | --- | --- | --- |
> > | 1 | 1.3741 | 0.728 | 30.40 | 30.48 |
> > | 2 | 0.8025 | 1.246 | 31.22 | 31.34 |
> > | 4 | 0.4876 | 2.051 | 32.87 | 33.04 |
> > | 8 | 0.3044 | 3.285 | 36.03 | 36.41 |
> >
> > ### **Q5: Reliability of entity prompting mechanism**
> >
> > Our goal is to provide explicit textual hints to guide the LLM decoder without introducing excessive computational overhead. Since existing LLMs are trained on general corpora, a domain gap exists for this task. As shown in the Table 6, mainstream MLLMs with few-shot prompting (e.g., InternVL2.5-1B, Qwen2.5VL-32B, GPT4o-mini) perform significantly worse than our method. Entity-based retrieval allows our approach to acquire task-specific prior knowledge efficiently without additional training. Furthermore, we manually verify the quality of entities on the Spot-the-Diff dataset: out of 300 samples, 24 contain noise (8%), indicating that the extracted entities are reliable.
> >
> > Table 6: results on CLEVR-Change dataset
> >
> > |  | B | M |  R | C |
> > | --- | --- | --- | --- | --- |
> > | SCORER(ICCV’23) | 56.3  | 41.2  | 74.5  | 126.8  |
> > | InterVL2.5-1B | 10.7 | 14.0 | 30.6 | 22.6 |
> > | Qwen2.5VL-32B | 53.4 | 40.1 | 71.5 | 113.8 |
> > | GPT4o-mini | 22.8 | 31.7 | 62.3 | 57.7 |
> > | Ours | **59.5** | **41.4** | **78.1** | **154.7** |
> >
> > Through ablation studies, we further validate the effectiveness of the EPC module. Experimental results show that adding the EPC module improves the model’s performance on the Spot-the-diff dataset, demonstrating its effectiveness in enhancing change detection and captioning.
> >
> > Table 7: results on Spot-the-diff dataset
> >
> > | Entities | B | M | R | C |
> > | --- | --- | --- | --- | --- |
> > | 0 | 12.0  | 13.2  | 34.1  | 62.4  |
> > | 1 | 12.1 | 15.8 | 35.9 | 61.9 |
> > | 2 | 10.5 | 14.9 | 34.5 | 62.3 |
> > | 3 | 11.8 | 16.2 | 33.8 | 60.2 |
> > | 4 | 11.2 | 15.3 | 35.2 | 61.0 |
> >
> > At present, the extracted entities are incorporated into the model via direct prompt concatenation. While this approach is simple and effective, it may not fully exploit the fine-grained structure of entity information, and in such cases, the LLM may not fully utilize this prior knowledge during inference. More advanced fusion strategies could potentially yield better robustness. For example, instead of concatenating entity tokens directly, one could encode entities with a dedicated encoder and inject them into the decoder through cross-attention, allowing the model to selectively attend to entity features rather than treating them as flat text. Another possibility is to use a gating module that dynamically fuses textual information. We will include representative failure cases and a detailed analysis in the revised version.
> >
> > ### **Q6: Why multiple entities degrade performance**
> >
> > As shown in Table 7 adding entities generally improves performance compared to not using any, and the model is relatively insensitive to the number of entities. In the revised version, we will include a visualization illustrating how such conflicts affect the model’s attention and predictions.

---

> ### Author Response · Authors · 2025-11-25
> **response to reviewer ad1p**
>
> ### **Q7: Detailed comparison with FINER-MLLM**
>
> We provide a detailed conceptual and empirical comparison with FINER-MLLM below:
>
> 1. **Retrieval Method**
> - **FINER-MLLM**: Retrieves complete sentences from the training set based on global image pair similarity
> - **DAVIR (Ours)**: Extracts fine-grained entity-level noun phrases (e.g., "blue car", "parking lot") that correspond to specific changed objects
>
> Entity-level guidance is more precise and less prone to introducing irrelevant context. Full sentences may contain information about unchanged regions, potentially confusing the model.
>
> 2. **Retrieved Textual Guidance**
> - **FINER-MLLM**: Implicitly regularizes the LLM's output probability distribution using retrieved tokens (Eq. 9 in their paper: `p_c = p_v(1 + p_r)`)
> - **DAVIR (Ours)**: Explicitly constructs entity-centric instructions that prompt the decoder (e.g., "[Prompt] + [Entity:] blue car, parking lot")
>
> In comparison, our method serves as a soft guidance. Unlike FINER, which directly modify the LLM’s final projection outputs, our approach is less dependent on the retrieval results and offers a more flexible way to incorporate entity-level priors.
>
> 3. **Visual Attention Supervision**
> - **FINER-MLLM**: Relies on dual constraints (intra-image independence + inter-image alignment) applied to Q-Former features
> - **DAVIR (Ours)**: Directly supervises the ViT encoder's final-layer attention maps using KL divergence with SAM-generated change masks
>
> Explicit attention supervision with spatial masks ensures the encoder focuses on change regions during feature extraction, while  constraints on query features in Qformer operate at a more abstract level.
>
> 4. **Performance Comparison on Spot-the-Diff, IER, and CLEVR**
>
> Beyond the conceptual differences discussed earlier, we conduct a performance comparison across three datasets.
>
> Table 8: results on three datasets
>
> | Dataset | Method | BLUE-4 | METERO | ROUGE | CIDEr |
> | --- | --- | --- | --- | --- | --- |
> | **Spot-the-Diff** | FINER-MLLM | **12.6** | 13.5 | 34.7 | 59.9 |
> |  | DAVIR (Ours) | 12.1 | **15.8** | **35.9** | **61.9** |
> | **IER** | FINER-MLLM | 12.9 | 15.1 | 40.4 | 52.1 |
> |  | DAVIR (Ours) | **16.7** | **16.2** | **41.2** | **65.2** |
> | **CLEVR** | FINER-MLLM | 55.6 | 36.6 | 72.5 | 137.2 |
> |  | DAVIR (Ours) | **59.5** | **41.4** | **78.1** | **154.7** |
>
> Across the three datasets, DAVIR outperforms FINER-MLLM. On Spot-the-Diff, it achieves notable gains in M (+2.3), C (+2.0), indicating better semantic understanding of tiny changes. On IER, DAVIR shows larger improvements across all metrics, with particularly strong gains in B (+3.8) and C (+13.1), highlighting its effectiveness on diverse global changes. On CLEVR-Change, DAVIR achieves the highest scores across all metrics, including B (+3.9), M (+4.8), R (+5.6), and C (+17.5), demonstrating its robustness and strong performance on synthetic datasets with significant viewpoint variations. Overall, these results indicate that DAVIR is especially effective at capturing object-level changes.
>
> ### **Q8: Applicability beyond Image Change Captioning**
>
> To evaluate the extensibility of our proposed components beyond image change captioning, we conduct additional experiments on the image change detection task. Specifically, we use ChangeAgent as the baseline model, integrate our FAC module into its vision encoder, and test on the LEVIR-MCI dataset, using mIoU as the primary evaluation metric. The results show that this modification consistently improves the model’s performance on the change detection task, demonstrating that our mechanism is not limited to ICC but can also benefit other visual difference reasoning tasks.
>
> Table 9: results on LEVIR-MCI datasets
>
> | Method | mIoU |
> | --- | --- |
> | ACABFNet  | 84.43 |
> | DARNet  | 84.99 |
> | DMINet | 85.37 |
> | BiFA | 85.68 |
> | Change-Agent | 86.43 |
> | Change-Agent w_FAC | **86.74** |
>
> These findings indicate that our method has broader utility and can be transferred to related domains involving visual difference reasoning.

---

### Official Review · Reviewer_BEpt · 2025-10-31

**Soundness:** 3
**Presentation:** 2
**Contribution:** 2
**Rating:** 4
**Confidence:** 4

**Summary:**

This paper tackles the Image Change Captioning (ICC) task by proposing DAVIR, a visiolinguistic regularization framework designed to enhance Multi-modal LLMs. DAVIR introduces two key modules: (1) Fine-Grained Attention Control (FAC) to align encoder attention with change masks, and (2) Entity Prompt Construction (EPC) to guide the decoder with relevant textual entities. Experimental results on three benchmarks demonstrate consistent performance gains.

**Strengths:**

1.	The problem this paper focuses on is reasonable and worth exploring.
2.	Experiments on three diverse datasets demonstrate consistent improvements.

**Weaknesses:**

1.	Change Mask Extraction. The method generates change regions via pixel-wise subtraction and subsequent clustering. This approach, however, is plagued by pseudo-changes (e.g., viewpoint or illumination variations).
2.	Entity Prompt Construction may inadvertently worsen hallucinations. While designed to mitigate them, if CLIP misidentifies entities, it can induce or reinforce hallucinations in the LLM. Furthermore, CLIP itself can exhibit more hallucinations than powerful MLLMs like InstructBLIP, making it unclear whether this component is actually beneficial.

**Questions:**

1.	Effectiveness of Change Mask Extraction on CLEVR-DC

---

> ### Author Response · Authors · 2025-11-25
> **response to reviewer BEpt**
>
> ### Q1: **Masks Sensitive to Pseudo-Changes**
>
> The Spot-the-Diff dataset used in our experiments consists of surveillance images captured at different times, which may include illumination and viewpoint variations. In such scenarios, pixel-wise subtraction remains a generally effective and adaptable strategy. By contrast, CLEVR-DC is a synthetic dataset where pseudo-changes caused by extreme viewpoint shifts seldom occur in real-world settings. Even under these artificially challenging conditions, our method demonstrates strong robustness. Importantly, our primary objective is to guide the visual encoder to perceive objects rather than focus solely on change regions, so the extracted masks do not need to be perfectly accurate. As shown in Table 1, initial experiments with BLIP indicate that even coarse object masks (generated by applying SAM to segment all objects in the image) can improve model performance, reflecting enhanced reasoning capability.
>
> Table 1: results on Spot-the-diff dataset
>
> |  | B | M | R | C |
> | --- | --- | --- | --- | --- |
> | baseline | 11.5 | 13.0 | 33.7 | 60.4 |
> | w_FAC (coarse masks) | 11.8 | 12.8 | 34.0 | 60.8 |
> | w_FAC | **12.0** | **13.2** | **34.1** | **62.4** |
>
> To further validate robustness, we evaluate our method on two synthetic datasets—CLEVR-Change and CLEVR-DC—that simulate scenarios seldom encountered in real-world applications.
>
> As shown in Tables 2 and 3, the consistent improvements indicate that our FAC module effectively mitigates pseudo-changes by directing attention toward semantic objects instead of being misled by low-level pixel differences.
>
> Table 2: results on CLEVR-Change dataset
>
> |  | B | M | R | C |
> | --- | --- | --- | --- | --- |
> | baseline | 57.7 | **41.7** | 74.7 | 129.6 |
> | +FAC | **59.5** | 41.4 | **78.1** | **154.7** |
>
> Table 3: results on CLEVR-DC dataset
>
> |  | B | M | R | C |
> | --- | --- | --- | --- | --- |
> | baseline | 62.6 | 34.8 | 69.6 | 105.0 |
> | +FAC | **62.5** | **35.7** | **71.1** | **111.5** |

---

> > ### Author Response · Authors · 2025-11-25
> > **response to reviewer BEpt**
> >
> > ### Q2:  **Entity Prompts May Amplify Hallucinations**
> >
> > Our goal is to provide explicit textual cues that can guide the LLM decoder while avoiding unnecessary computational overhead. Because current LLMs are trained on broad, general-purpose corpora, there remains a domain gap for this specialized task. As shown in Table 4, mainstream MLLMs with few-shot prompting (e.g., InternVL2.5-1B, Qwen2.5VL-32B, GPT-4o-mini) perform substantially worse than our method. By contrast, entity-based retrieval enables our approach to efficiently acquire task-specific prior knowledge without requiring additional training. Moreover, we manually assess the entity quality on the Spot-the-Diff dataset: among 300 samples, only 24 contain noise (8%), demonstrating that the extracted entities are reliable.
> >
> > Table 4: results on CLEVR-Change dataset
> >
> > |  | B | M |  R | C |
> > | --- | --- | --- | --- | --- |
> > | SCORER(ICCV’23) | 56.3  | 41.2  | 74.5  | 126.8  |
> > | InterVL2.5-1B | 10.7 | 14.0 | 30.6 | 22.6 |
> > | Qwen2.5VL-32B | 53.4 | 40.1 | 71.5 | 113.8 |
> > | GPT4o-mini | 22.8 | 31.7 | 62.3 | 57.7 |
> > | Ours | **59.5** | **41.4** | **78.1** | **154.7** |
> >
> > Through ablation studies, we further validate the effectiveness of the EPC module as shown in Table 5. Experimental results show that adding the EPC module improves the model’s performance on the Spot-the-diff dataset, demonstrating its effectiveness in enhancing change detection and captioning.
> >
> > Table 5: results on Spot-the-diff dataset
> >
> > | Entities | B | M | R | C |
> > | --- | --- | --- | --- | --- |
> > | 0 | 12.0  | 13.2  | 34.1  | 62.4  |
> > | 1 | 12.1 | 15.8 | 35.9 | 61.9 |
> > | 2 | 10.5 | 14.9 | 34.5 | 62.3 |
> > | 3 | 11.8 | 16.2 | 33.8 | 60.2 |
> > | 4 | 11.2 | 15.3 | 35.2 | 61.0 |
> >
> > When training captions contain noise, our entity bank is constructed solely from entities extracted from ground-truth captions in the training set. This closed-set design ensures that even if CLIP retrieves suboptimal entities, they remain semantically aligned with the dataset domain rather than entirely irrelevant concepts. To avoid any possibility of data leakage, we compute the cross-split 4-gram overlap between retrieved captions and ground-truth captions in the test sets to assess redundancy. As shown in Table 6, there is a substantial gap between training and test entities, confirming that our closed-set entity bank does not leak information from the test set.This closed-set constraint ensures that when CLIP retrieves suboptimal entities, they are still semantically relevant to the dataset domain rather than entirely unrelated concepts. the averaged 4-gram overlaps are 6.1% for the two synthetic datasets and 0.45% for the two real-world datasets. The relatively higher overlaps in the synthetic datasets can be attributed to their template-based caption generation, where fixed linguistic patterns are repeatedly filled with varying object or attribute tokens, leading to partial phrase reuse across splits. In contrast, the real-world datasets are human-annotated, resulting in more diverse linguistic expressions and much lower overlap. Overall, there is a large gap between training and test samples, confirming that our closed-set memory bank does not leak information from the test set.
> >
> > Table 6: 4-Gram Overlap Between Retrieved and Test Set Ground-Truth Captions
> >
> > | CLEVR-Change | Spot-the-diff | Image-editing-request | CLEVR-DC |
> > | --- | --- | --- | --- |
> > | 5.55% | 0.56% | 0.33% | 6.75% |

---

### Official Review · Reviewer_CUgz · 2025-11-01

**Soundness:** 3
**Presentation:** 3
**Contribution:** 3
**Rating:** 6
**Confidence:** 3

**Summary:**

In this paper, the authors focus on the task of image change captioning. Existing methods rely solely on caption-level supervision, a limitation that may lead to the oversight of fine-grained changes. To address this, the authors propose a difference-aware vision-language regularization framework, which regularizes the MLLM using visual change masks and textual change entities. Specifically, their proposed fine-grained attention control module extracts change masks for each image; these masks are then used to regularize the self-attention maps of the MLLM’s encoder. Experimental results demonstrate the effectiveness of the proposed method.

**Strengths:**

1. The proposed method incorporates visual change masks to regularize the fine-tuning of the MLLM’s encoder and capture subtle changes.
2. The entity-prompt construction scheme is designed to regularize the MLLM’s decoder, thereby enabling more accurate image change caption generation.
3. Experimental results demonstrate the effectiveness of the proposed method.

**Weaknesses:**

1. The derivation of change masks relies primarily on pixel-level discrepancies, but there are cases where the views of the two images are not closely aligned. In such scenarios, the derived change masks may be inaccurate, thereby impacting subsequent feature extraction and change caption generation.
2. In the entity-guided prompting strategy, the extracted entities largely rely on the SAM model, which may perform poorly in scenarios where SAM fails to detect objects successfully.

**Questions:**

As listed above.

---

> ### Author Response · Authors · 2025-11-25
> **response to reviewer CUgz**
>
> Thank you very much for your valuable suggestions. Below, we provide our responses to the questions you raised.
>
> ### Q1: **Sensitivity of Masks Under Viewpoint Misalignment**
>
> The datasets Spot-the-diff used in our experiments consists of  surveillance images captured at different times, containing potential illumination and viewpoint variations. In such cases, pixel-wise subtraction remains a generalizable and adapted strategy. In contrast, CLEVR-DC is a synthetic dataset where pseudo-changes caused by extreme viewpoint shifts rarely occur in real-world scenarios. Even under these artificially challenging settings, our method still demonstrates strong robustness. Importantly, our main goal is to guide the visual encoder to perceive objects rather than focus on change regions, so the extracted masks do not need to be perfectly precise. In initial experiments with BLIP as shown in Table 1, we observed that even coarse object masks (generated by applying SAM to segment all objects in the image) can improve model performance, indicating enhanced reasoning capability.
>
> Table 1: results on Spot-the-diff dataset
>
> |  | B | M | R | C |
> | --- | --- | --- | --- | --- |
> | baseline | 11.5 | 13.0 | 33.7 | 60.4 |
> | w_FAC (seg all objects) | 11.8 | 12.8 | 34.0 | 60.8 |
> | w_FAC | **12.0** | **13.2** | **34.1** | **62.4** |
>
> To further verify robustness, we evaluate our method on two synthetic datasets—CLEVR-Change and CLEVR-DC—which simulate scenarios rarely encountered in real-world applications.
>
> As shown in Table 2 and Table 3, the consistent improvements demonstrate that our FAC module effectively handles pseudo-changes by regularizing attention toward semantic objects rather than being misled by low-level pixel differences.
>
> Table 2: results on CLEVR-Change dataset
>
> |  | B | M | R | C |
> | --- | --- | --- | --- | --- |
> | baseline | 57.7 | **41.7** | 74.7 | 129.6 |
> | +FAC | **59.5** | 41.4 | **78.1** | **154.7** |
>
> Table 3: results on CLEVR-DC dataset
>
> |  | B | M | R | C |
> | --- | --- | --- | --- | --- |
> | baseline | 62.6 | 34.8 | 69.6 | 105.0 |
> | +FAC | **62.5** | **35.7** | **71.1** | **111.5** |
>
> ### Q2:  **Dependency on SAM Accuracy for Reliable Entity Extraction**
>
> We clarify how our method handles SAM segmentation failure cases. Our entity prompt construction is designed with multiple fallback mechanisms. First, we extract entities from both before and after images. If SAM cannot generate a mask for one image, entities can still be extracted from the other. If SAM cannot generate masks for both images, the returned prompt is empty, and the model naturally falls back to caption-level supervision only.
>
> As shown in Table 4 (row 2) in the submitted version, our model without EPC still achieves strong performance, demonstrating that the FAC module alone provides robust change understanding. When SAM segmentation contains errors, our explicit prompting strategy mitigates noise. We use the phrasing “changed objects may contain entities” rather than “changed objects are entities.” This soft guidance encourages the decoder to consider these entities as candidates without forcing them into the caption. LLM’s reasoning capability helps filter out irrelevant entities that do not align with actual visual changes observed in the encoded features.

---

### Official Review · Reviewer_Hdfj · 2025-11-03

**Soundness:** 2
**Presentation:** 2
**Contribution:** 2
**Rating:** 2
**Confidence:** 4

**Summary:**

This paper proposes a framework built upon InstructBLIP to enhance the description of image changes. The main contributions are two techniques: Fine-grained Attention Control (FAC) and Entity Prompt Construction (EPC). FAC directs the model’s attention to the regions where changes occur, while EPC improves the quality of change-aware caption generation. The proposed method demonstrates strong performance across three benchmark datasets, and the authors conduct comprehensive ablation studies to analyze the contribution of each component.

**Strengths:**

1. The motivation of the paper is clearly stated.
2. The authors evaluate their method on multiple datasets, ensuring a comprehensive comparison.
3. Across all datasets, the proposed approach consistently outperforms or matches existing baselines.

**Weaknesses:**

1. In the ablation study, the performance trend is not consistent. For example, in Table 4, the configuration using both FAC and EPC does not achieve the best results. This suggests that the two proposed components might have conflicting effects.
2. The paper lacks analysis or explanation for the performance variations observed in Table 4, which makes it difficult to understand the interaction between FAC and EPC.
3. The selected baselines are somewhat outdated. Recent models such as Qwen-VL and InternVL have demonstrated strong performance on image change captioning (ICC) tasks. Including results from these frontier open-source models would make the comparison more comprehensive.
4. The ablation study focuses mainly on the Spot-the-Diff dataset, which primarily contains object-level changes. This dataset alone is not representative enough to fully evaluate the generalizability of the proposed approach.
5. A failure case analysis is missing. Providing examples of where the method fails could offer valuable insights into its limitations and potential areas for improvement.

**Questions:**

The paper only provides examples involving object-level changes. It remains unclear how the proposed method would handle non-object changes, such as color or style transformations (e.g., making the entire image hue red). It would be helpful if the authors could include such examples or discuss how their approach generalizes to these types of changes.

---

> ### Author Response · Authors · 2025-11-25
> **response to reviewer Hdfj**
>
> Thank you very much for your valuable suggestions. Below, we provide our responses to the questions you raised.
>
> ### Q1 & Q2: Inconsistent Ablation Results and Lack of Interaction Analysis
> The results demonstrate complementarity rather than conflict. When using FAC alone, the model achieves improvements across all metrics, especially BLEU 12.0 (+0.5) and CIDEr 62.4 (+2.0). When using EPC alone, the model improves on 4 out of 5 metrics, notably METEOR 16.4 (+3.4), ROUGE 35.2 (+1.5), and SPICE 22.9 (+1.3). These results clearly show that both FAC and EPC provide consistent gains over the baseline, each contributing from different perspectives. FAC strengthens spatial grounding and enhances the model’s ability to localize and describe visual differences, while EPC enriches the linguistic space by introducing a more diverse and semantically meaningful entity vocabulary.
>
> Combining FAC and EPC yields a more balanced performance and larger gains compared to the baseline. Overall, the full model achieves the best results across all metrics, confirming that FAC and EPC work synergistically.
>
> ### Q3: Outdated Baseline Selection
>
> We extensively experiment with frontier MLLMs (Qwen2.5-VL-32B, InternVL, GPT4o-mini) in a few-shot setting at the project’s inception, as shown in Table 1, which reveals a critical limitation: despite their impressive general-purpose reasoning abilities, current frontier multimodal LLMs struggle to maintain high performance on fine-grained change captioning, particularly on synthetic datasets with significant viewpoint variations and templated language, such as CLEVR-Change.
>
> To ensure a fair comparison with existing LLM-based change captioning approaches, we follow prior works and implement our method on the BLIP framework. Furthermore, our model is plug-and-play, as FAC and EPC can be seamlessly integrated into standard vision–language architectures.
>
> Table 1: results on CLEVR-Change dataset
>
> |  | B | M |  R | C |
> | --- | --- | --- | --- | --- |
> | SCORER(ICCV’23) | 56.3  | 41.2  | 74.5  | 126.8  |
> | InterVL2.5-1B | 10.7 | 14.0 | 30.6 | 22.6 |
> | Qwen2.5VL-32B | 53.4 | 40.1 | 71.5 | 113.8 |
> | GPT4o-mini | 22.8 | 31.7 | 62.3 | 57.7 |
> | Ours | **59.5** | **41.4** | **78.1** | **154.7** |
>
> ### Q4: Limited Dataset Diversity in Ablation Study
>
> Spot-the-diff is a widely used surveillance dataset in which the changed objects are typically small and captured under potential viewpoint shifts and illumination variations due to different capture times. As object-level changes are inherently more difficult to detect than global changes such as illumination variations, we choose Spot-the-diff as the primary dataset for our ablation studies. In addition to Spot-the-diff, we conduct ablation studies on the Image-Editing-Request (IER) dataset as shown in Table 2, which contains additional types of global changes, including human-induced contrast and brightness adjustments, global appearance shifts, and holistic image transformations. This provides a complementary evaluation setting to assess the generalizability of our approach.
>
> Table 2: results on Image-Editing-Request dataset
>
> |  | B | M | R | C |
> | --- | --- | --- | --- | --- |
> | baseline | 13.7 | 16.1 | 40.0 | 58.3 |
> | +FAC | **16.7** | **16.2** | **41.2** | **65.2** |
>
> Our FAC module achieves substantial improvements across all metrics, particularly in CIDEr (+6.9) and BLEU-4 (+3.0), demonstrating robust generalization. The fact that our method performs well on both types of changes—object-level on Spot-the-Diff and global on IER—strongly indicates its generalizability. Through qualitative analysis of the extracted masks on IER, we observe that our method consistently identifies objects as change regions, even in the presence of global contrast and brightness variations. This demonstrates that our approach is robust and can adapt to diverse change types by focusing on semantically meaningful regions.
>
> ### Q5: Failure Case Analysis
>
> At present, the extracted entities are incorporated into the model via direct prompt concatenation. While this approach is simple and effective, it may not fully exploit the fine-grained structure of entity information, and in such cases, the LLM may not fully utilize this prior knowledge during inference. More advanced fusion strategies could potentially yield better robustness. For example, instead of concatenating entity tokens directly, one could encode entities with a dedicated encoder and inject them into the decoder through cross-attention, allowing the model to selectively attend to entity features rather than treating them as flat text. Another possibility is to use a gating module that dynamically fuses textual information. We will include representative failure cases and a detailed analysis in the revised version.

---

### Author Response · Authors · 2025-11-29
**Summary of Response to AC**

Dear AC,

We seek to inform you that we have posted detailed responses to all reviewers. Our rebuttal focuses on **clarifying our design choices** and **providing additional experimental evidence** to address specific questions raised during the review process.

1. **Overview of Responses to Reviewers.** To address the constructive queries from Reviewers [**ad1p, BEpt, CUgz**], we have provided the following clarifications and supplementary results:
    - **Validation of Robustness:** We conducted additional experiments on the **IER** and synthetic datasets (**CLEVR-Change and CLEVR-DC**) to verify our method's performance under viewpoint shifts and non-object changes.
    - **Clarification of Methodology:** We provided detailed explanations regarding our attention supervision and entity prompting mechanisms to resolve ambiguities about our design rationale.
    - **Comprehensive Comparisons:** We included requested computational analyses (time/memory) and comparisons with related works like FINER-MLLM further substantiate our state-of-the-art performance.
    - **Generalizability Beyond ICC:** We provided additional discussion on model generalizability beyond image change captioning, including results on the **LEVIR-MCI** change detection benchmark to demonstrate applicability across related tasks.
2. **Specific Concern Regarding Reviewer Hdfj (Score: 2).** While Reviewer Hdfj acknowledged our sound motivation and explicitly stated that our method "consistently outperforms" baselines, they assigned a score of 2.

    The concerns raised were primarily regarding the *completeness* of the experimental analysis (e.g., requests for modern baselines and failure cases), rather than the validity of the method itself. In our response, we have fully supplied the requested data:

    - **Frontier Baselines:** We provided results against **Qwen2.5-VL, InternVL, and GPT4o-mini**, confirming our performance advantage.
    - **Additional Analysis:** We included the requested failure case analysis and synergy discussions.

Considering our responses to all reviewers and the specific evidence provided to address Reviewer Hdfj's concerns, we respectfully invite the AC to evaluate the paper's overall merit.

Sincerely,

The Authors

---

### Meta-Review · Area_Chair_igp2 · 2026-01-09

**Summary:**

This paper proposes a framework, DAVIR, built upon InstructBLIP to enhance the description of image changes. The main contributions are two techniques: Fine-grained Attention Control (FAC) and Entity Prompt Construction (EPC). FAC directs the model’s attention to the regions where changes occur, while EPC improves the quality of change-aware caption generation. Thanks to the design of the entity-prompt construction scheme to regularize the MLLM’s decoder, more accurate image change caption generation is enabled as demonstrated in the experiments.  The reviewers, however, raised concerns regarding insufficient analysis of the obtained results to clarify the effectiveness of the two introduced components.  More evaluations to clarify the ability/limitation of the proposed method were also pointed out as concerns. They include not object-level changes such as viewpoint and/or illumination changes, dependency of SAM accuracy, and computational analysis.  The authors tried to address the raised concerns in their rebuttal.  AC thinks that some were resolved while some were not fully resolved.

Regarding Reviewer Hdfj’ concerns on conflicting effects of two proposed components, the rebuttal stated that FAC and EPC are complementary but not conflicting by arguing FAC’s benefits to BLEU and CIDEr and EPC’s benefits to METEOR, ROUGE, and SPICE.  However, if we take a careful look at Table 4, AC finds that FAC indeed boosts CIDEr but not so much for BLEU (SPICE has slightly better benefit than BLEU).  AC also finds that EPC boosts METEOR while degrades CIDEr; ROUGE and SPICE are boosted marginally.  Benefits by FAC and EPC to BLEU are both quite limited.  It is thus more reasonable to conclude as follows: FAC contributes to CIDEr, and EPC does to METEOR.  EPC also contributes to ROUGE and SPICE marginally.  FAC and EPC fail to contributes BLEU.  EPC degrades CIDEr.  Based on this observation, AC agrees FAC’s and EPC’s complementarity to some extent while AC acknowledges that unclearness of the two component effects remain.  Why does EPC degrade CIDEr? Why do both fail to contribute to BLEU?  More rigorous analysis is required to accept the authors’ claim.

Regarding the concern on generalizability/robustness, the authors provided additional experiments on IER and CLEVER-change, CLEVER-DC, which should be appreciated.  However, those experiments are evaluation with +FAC only: EPC is dropped somehow (metric SPICE also).  Since Table 4 shows CIDEr is degraded by EPC, we cannot confirm DAVIR’s improvements precisely. With the provided results and arguments, it is still unclear whether the proposed method can be applied to scenarios beyond object-level changes.  The authors also admitted, in their rebuttal, limitation of robustness by addressing that “While this approach is simple and effective, it may not fully exploit the fine-grained structure of entity information, and in such cases, the LLM may not fully utilize this prior knowledge during inference.  More advanced fusion strategies could potentially yield better robustness.” This means that robustness the proposed method is limited.

Concerns regarding on dependency on SAM accuracy and computational analysis are well addressed.

AC thinks that the task this paper focuses on is reasonable and worth exploring, and the proposed method is interesting; however, more thorough analysis of the introduced FAC and EPC, in particular, their relation in effectiveness, should be required.  The case of not object-level changes should be deeply evaluated and argued to make the paper solid.  AC thinks that the remaining concerns outweigh the technical contribution of this paper; the paper will be beneficial from substantial improvement.  For this, this paper cannot be accepted.

**Reviewer Concerns:**

Main concerns are on insufficient analysis of the obtained results to clarify the effectiveness of the two introduced components, insufficient evaluations including not object-level changes such as viewpoint and/or illumination changes, dependency of SAM accuracy, and computational analysis.  As written above, concerns regarding dependency of SAM accuracy and computational analysis were well addressed while the other crucial concerns were not resolved.

**Reviewer Scores:**

Reviewer Hdfj would keep the initial score or raise the score to 4 optimistically because concerns on conflicting effects of two proposed components, and on generalizability to not object-level changes are not fully addressed.  Reviewers CUgz and BEpt would keep their initial scores because generalizability/robustness are not fully addressed.  Reviewer ad1p would keep his/her initial score as well because of the same reason as Reviewers CUgz and BEpt.

---

### Decision · Program_Chairs · 2026-01-26

Reject